# Design and Performance Analysis of 32 × 32 Memory Array SRAM for Low-Power Applications

Xingsi Xue [1], Aruru Sai Kumar [2,*], Osamah Ibrahim Khalaf [3], Rajendra Prasad Somineni [2], Ghaida Muttashar Abdulsahib [4], Anumala Sujith [2], Thanniru Dhanuja [2] and Muddasani Venkata Sai Vinay [2]

1 Fujian Provincial Key Laboratory of Big Data Mining and Applications, Fujian University of Technology, Fuzhou 350011, China
2 Department of ECE, VNR Vignana Jyothi Institute of Engineering and Technology, Hyderabad 500090, India
3 Department of Solar, Al-Nahrain Research Center for Renewable Energy, Al-Nahrain University, Jadriya, Baghdad 64040, Iraq
4 Department of Computer Engineering, University of Technology, Baghdad 10066, Iraq
* Correspondence: saikumar_a@vnrvjiet.in

**Abstract:** Computer memory comprises temporarily or permanently stored data and instructions, which are utilized in electronic digital computers. The opposite of serial access memory is Random Access Memory (RAM), where the memory is accessed immediately for both reading and writing operations. There has been a vast technological improvement, which has led to tremendous information on the amount of complexity that can be designed on a single chip. Small feature sizes, low power requirements, low costs, and great performance have emerged as the essential attributes of any electronic component. Designers have been forced into the sub-micron realm for all these reasons, which places the leakage characteristics front and centre. Many electrical parts, especially digital ones, are made to store data, emphasising the need for memory. The largest factor in the power consumption of SRAM is the leakage current. In this article, a 1 KB memory array was created using CMOS technology and a supply voltage of 0.6 volts employing a 1-bit 6T SRAM cell. We developed this SRAM with a 1-bit, 32- × 1-bit, and 32 × 32 configuration. The array structure was implemented using a 6T SRAM cell with a minimum leakage current of 18.65 pA and an average delay of 19 ns. The array structure was implemented using a 6T SRAM cell with a power consumption of 48.22 µW and 385 µW for read and write operations. The proposed 32 × 32 memory array SRAM performed better than the existing 8T SRAM and 7T SRAM in terms of power consumption for read and write operations. Using the Cadence Virtuoso tool (Version IC6.1.8-64b.500.14) and 22 nm technology, the functionality of a 1 KB SRAM array was verified.

**Keywords:** 6T SRAM cell; CMOS; decoder; read and write access delay; power consumption

## 1. Introduction

In today's world, satellite communication applications are widely used. They are used for disaster monitoring, broadcasting, military operations, and other purposes. Lightweight satellites are being produced as technology progresses to lower the cost of production and maintenance [1–4]. Lightweight satellites need high-density memory cells because of their small size. Due to their high packing density and superior logic performance in digital data processing and the satellite's control system, SRAM cells are an excellent choice for this application. High-energy charged particles are present in space. When a particle strikes a logic device, electron–hole pairs are created. These electron–hole pairs are split apart by the electric field and collected at the sensitive node. Charge build-up causes a brief voltage pulse to be generated [5]. In biomedical applications, high-density memory is crucial, as are electronic devices. The major reason for running memory at low voltage is to extend the battery life and use as little energy as possible. The typical 6T SRAM cell has a low level of read process noise immunity. As the supply voltage drops, the noise immunity

becomes much lower. Therefore, ordinary 6T SRAM cannot function at low supply voltages. By segregating bit lines from the storage node, decoupled 7T and 8T SRAM cells offer increased noise immunity in reading mode; however, those cells have significant leakage power. Although SRAM memory comprises millions of SRAM cells in standby mode, an exponential rise in the leakage power increases the overall power consumption of SRAM memory [6–10].

Modern very-large-scale integration (VLSI) systems have improved the organisation of embedded memory. Static random access memory (SRAM) and dynamic random access memory are the two main types of random access memory cells (DRAM). While capacitors and a single transistor are used to create DRAM cells, The term "static" indicates that the circuit is free of the floating node situation and that, at any given moment, all of its components either are linked to $V_{dd}$ or $V_{SS}$. The term "random" indicates that the data can be retrieved at any moment and from any location, regardless of the arrangement of the data. Accessing involves memory search and bit storing. One bit is stored in each cell [11–13].

Transistors and latches are used in the construction of SRAM cells. As a result, charging and discharging the capacitors used for data storage and retrieval takes a long time and much power. Because of this advantage, SRAM cells are frequently used in systems on a chip (SoCs) [14–17], where they form a vital part of the design and installation. Numerous SRAM cell designs have been introduced, which are made for exceptional performance in response to the growing demand for power reduction and increased productivity in contemporary SoC technologies. In contrast, the 6T SRAM cell is usually regarded as offering a superb balance between size and performance. Larger SRAM arrays, which are frequently utilised in AI and IoT devices, are employed by SoC technologies to enhance performance [18–20]. Therefore, the consequent increase in the chip size, price, and power consumption is caused by the area influence of expanding the SRAM configuration on the chip.

As the transistor channel size of very-large-scale silicon chips is continuing to shrink, there are more high-speed-capable devices. Because of the reducing size of transistors with each new generation, bulk CMOS technology has enabled current integrated electronics to operate at a continually higher speed. The development of bulk CMOS faces considerable challenges because of the inherent material and computational limitations of the technology. The 1-bit 6T SRAM cells, write driver circuits, sense amplifiers, row and column decoders, and other peripherals are used to build SRAM arrays [21]. The proposed design optimizes the process through the following parameters.

The major novel contributions of the proposed method are as follows:

- A 1 KB memory array was created through CMOS technology, which reduces the power consumption for the read and write operations.
- The 6T RAM was designed using a 32 × 32 memory (1 KB) array by considering reduced leakage current and low power consumption for read and write operations of a 32 × 32 SRAM memory array.
- The experimental results of the proposed research paper were verified on CMOS 22 nm technology using the Cadence Virtuoso tool (Version IC6.1.8-64b.500.14).

The organization of the paper is as follows. Section 2 represents the literature work and the various techniques. Section 3 deals with the SRAM array architecture. Section 4 provides the proposed 32 × 32 SRAM array architecture, followed by the analysis of the results in Section 5. The conclusion of the research work is discussed in Section 6.

## 2. Related Works

Jayram Shrivas et al. [22] discussed 7T SRAM bit cells that use a sleep method to cut down on SRAM leakage power. Instead of silicon dioxide, a high-k gate dielectric material based on silicon dioxide ($SiO_2$) was employed. The bit line pair utilized for the writing operations uses less discharge power thanks to the 7T cell. A high-threshold-voltage PMOS transistor employed as an additional sleep transistor was used. The width of the sleep

transistor has an inverse relationship with the voltage across the SRAM bit cell leaking, and the wake-up transistor is parallel-connected to the sleep transistor, thereby decreasing the sleep latency.

Shalini Singh et al. [23] implemented a 1KB memory with SRAM for data storage. The array structure was implemented using a 7T SRAM cell with a minimum leakage current of 20.16 pA and an average latency of 21 ns. This was accomplished using a 2D array made of the SRAM's fundamental unit cells. The Cadence Virtuoso tool was used to create the $32 \times 32$ array and other supporting components such as the address decoder, precharge circuit, write driver, and sense amplifiers in 45 nm technology. The power used by the read and write operations on the 1 KB SRAM-based memory differed significantly, at 51.57 mW and 447.3 mW, respectively. The noise margin, PVT fluctuations, and cell stability can all be impacted by lowering the nominal voltage.

One of the primary issues in SRAM design is the read stability and periodic precharge during the read/write cycle. The novel SRAM architecture created by Alex Gong et al. [24], with a focus on the reading operation in particular, was explained in this study as a solution to these two issues. Cell node inversion was made possible by the use of sense-amplifying cells. To increase the read resilience, it was proposed to separate the digital output bits from the data retention components in a read-SNM-free SRAM cell. This was created using the Cadence design software in 0.18 μm CMOS technology. Under the same operating conditions, this SRAM showed a decrease in total power consumption comparable to its traditional counterpart. In comparison to 6T SRAM cells, this approach requires eight transistors for each cell, resulting in an over 30% increase in the SRAM area.

Rashmi Bisht et al. [25] developed an SRAM array, and peripheral components such as the row decoder, precharge circuit, write driver circuit, sense amplifier, and column decoder were constructed and assembled. Differential-type sense amplifiers were used to lessen the noise because they can reject the voltage that has a common mode. To minimise the static power dissipation, they employed cross-coupled CMOS inverters. Strong noise immunity and low operating voltage are further benefits of this design. At low supply voltages, it was demonstrated that full CMOS SRAM cells are more stable than resistive load SRAM cells. The SRAM array's overall measured power consumption was 24.58 mW. For the design, the widely known gpdk180 library was utilised (or 180 nm technology node).

Himanshu Banga et al. [26] demonstrated a $16 \times 16$ SRAM with a reduced die area and leakage power. The SRAM's performance was enhanced as a result. Relaxing the space restrictions, the smaller peripheral used less energy. If the area of the SRAM memory is not a concern, then this method is useful for low-energy applications. Additionally, the sleep and forced transistor techniques were used in this study to further cut down the power usage. It was discovered that the forced transistor was 99.94% quicker than the sleep approach, while the forced transistor decreased the overall power usage by 56.92%. Using the Cadence tool, the $16 \times 16$ SRAM memory was developed, built, and analysed in the standard UMC 180 nm technology library.

To address the issues with the normal SRAM cell while enhancing the performance and power consumption, Shyam Akashe et al. [27] developed a read static noise margin-free SRAM cell with five transistors. When creating this unique cell, they made use of the memory value stream of ordinary programmes' significant bias toward zero at the bit level. The main findings that guided the design were that the node where the transistor is off is where the cell leakage is measured. Under the same design standards, the recommended cell area was 21.66% smaller than the 6T SRAM cell with a 28.57% speed improvement. The durability of the proposed cell, which simulates 45 nm technology, was computed utilising appropriate read/write operations. Additionally, the latency of the new cell was 70% lower than that of an SRAM cell with six transistors. Despite the fact that the leakage current doubled for every 10 degrees Celsius increase in temperature, the suggested cell had a memory cell access leakage current that was 72.10% less than that of the 6T SRAM cell.

### 3. SRAM Memory Array Architecture

The organisational framework of the SRAM memory array construction is depicted in Figure 1. There are two approaches to construct SRAM arrays, depending on the specifications: bit-oriented structure or word-oriented structure. In a bit-oriented structure, each position allows accessing a single bit of data. In contrast, the word-oriented design of SRAM maps each address to a word with n data bits. For the word-oriented layout, one sense amplifier is to be distributed over two, four, or more columns using a column decoder or column mux, which is addressed by the "K" address bits. One column shares a single sense amplifier in the bit-oriented organisation architecture, enabling a more detailed read operation. A 1-bit 6T SRAM cell, write driver circuits, precharge circuits, sense amplifiers, row decoders, and column decoders made up the SRAM array. The precharge circuit was utilised in the memory array configurations to equalise the bit lines before the operation mode. Figure 2 depicts the SRAM memory array implementation [28–32].

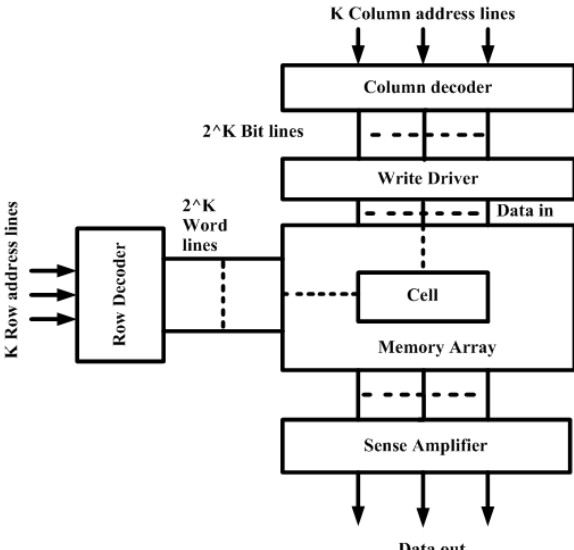

**Figure 1.** Structure of SRAM array.

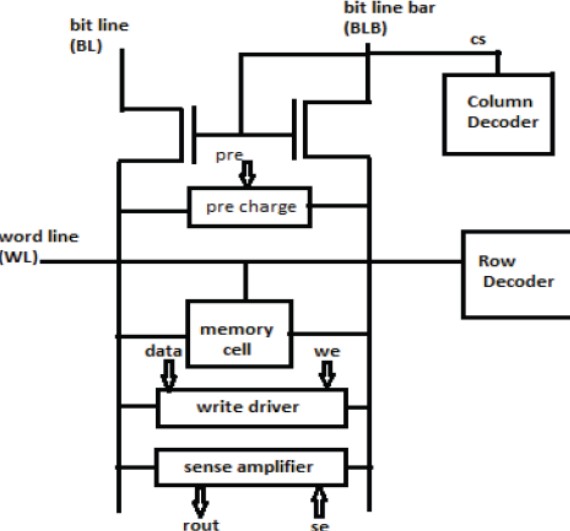

**Figure 2.** Implementation of SRAM array.

## 4. Proposed 32 × 32 SRAM Array Architecture

Using a 32 × 32 array, 1024 6T SRAM cells with 32 bits of output were used to create a 1 KB memory. An array of the form $2^K$ × m has $K$ address lines and $m$ data bits. Here, a $2^5$ × 32 array was utilised, with a 5:32 decoder because there are five address lines in the array. The implementation of this memory is shown in Figure 3. The Write enable (WE) signal activates the write driver, which drives the bit line from the precharge level to the ground with full-swing discharge. The write operation is unaffected by the order in which the word line and write drivers are enabled. The amount of rows and columns that make up a decoder's output are often word lines, whereas the column lines that make up a column are known as bit lines. At the point where the row and column lines connect, each SRAM cell is put into its own cell. The static random access memory (SRAM) used in this suggested research was a six-transistor SRAM. The decoder's output line swings high in response to the address input selection, picking the pass transistor for that particular row, which selects all of the cells in that specific row. The two data bits are B and Bbar lines. If one is high and the other is low, the cells from which we want to obtain the data are chosen.

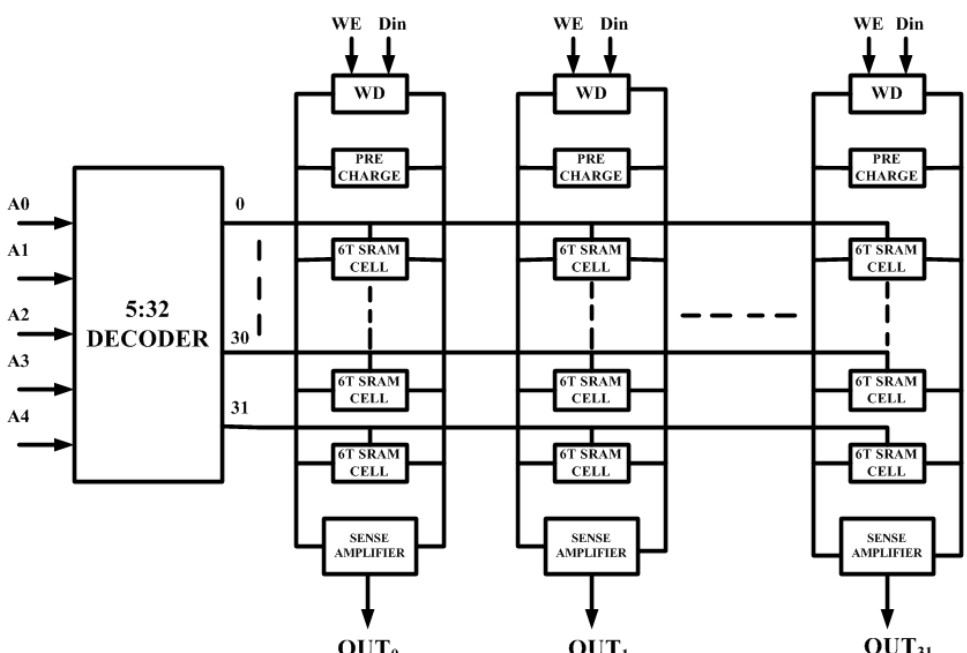

**Figure 3.** Architecture of the proposed 1 KB (32 × 32) SRAM memory.

### 4.1. 6T SRAM Bit Cell

An SRAM bit cell is composed of two inverters connected back to back to form a latch and two access transistors, as shown in Figure 4. This latch serves as a memory function because, when the input is "1" on one side of the latch, it becomes "0" on the other side. This memory part is connected to the $BL$ and $\overline{BL}$ to read and write. To access these lines, we used access transistors. The word line (WL) is connected to the access transistors.

If the WL is equal to "1", then both access transistors are in the ON state and the $BL$ and $\overline{BL}$ can be accessed. This is helpful for the reading and writing operations. If the WL is equal to "0", the access is lost, and the memory will be in the HOLD state. In this case, the $BL$ and $\overline{BL}$ serve as the output lines when we need to read from the memory and as the input lines when we need to write to it. Capacitors that are precharged were utilised for reading and writing the data.

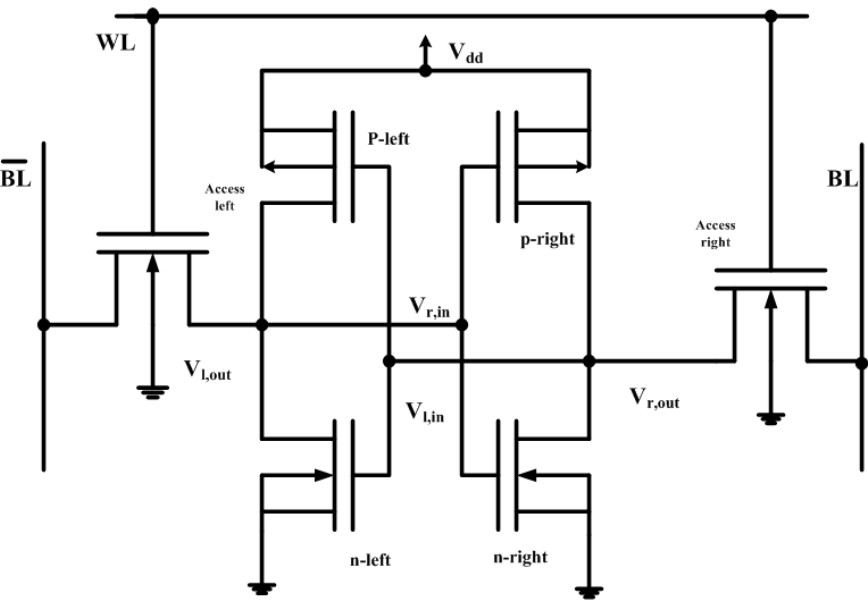

**Figure 4.** 6T SRAM bit cell.

### 4.2. Precharge Circuit

A *BL* and $\overline{BL}$ pair, also known as a bit line pair (*BL*), is present in a 6T SRAM cell, as shown in Figure 5. A PMOS is connected to the precharge circuit at the end of each bit line. When the precharge input is set to 0, this circuit will use an equaliser transistor to precharge the bit lines to $V_{dd}$ and perfectly equalise them. A precharge and equaliser circuit consists of three PMOS transistors, and bit lines are connected to $V_{dd}$ when the transistors are in the ON state, i.e., the precharge is active-low.

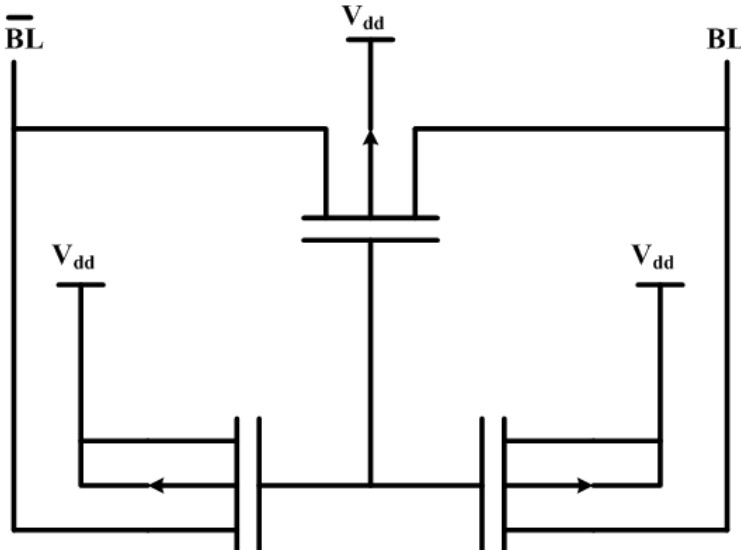

**Figure 5.** Precharge circuit.

### 4.3. Sense Amplifier

The sense amplifier (SA), which is shown in Figure 6, amplifies a small analogue differential voltage that forms on the bit lines during a read access cycle. A single-ended, full-swing digital output is produced by the amplifier. The SRAM cell is smaller when the SA is used because the driver transistors do not have to fully discharge the bit lines. The SA must follow precise power requirements in order to function properly. To begin, the

bare minimum discrepancy voltage that the SRAM cell generates across the bit lines should be less than the minimum discrepancy voltage swing that is required at the SA's input. Second, the SA should be sufficient to provide the output within the sense amplification time after receiving a minimal input differential voltage.

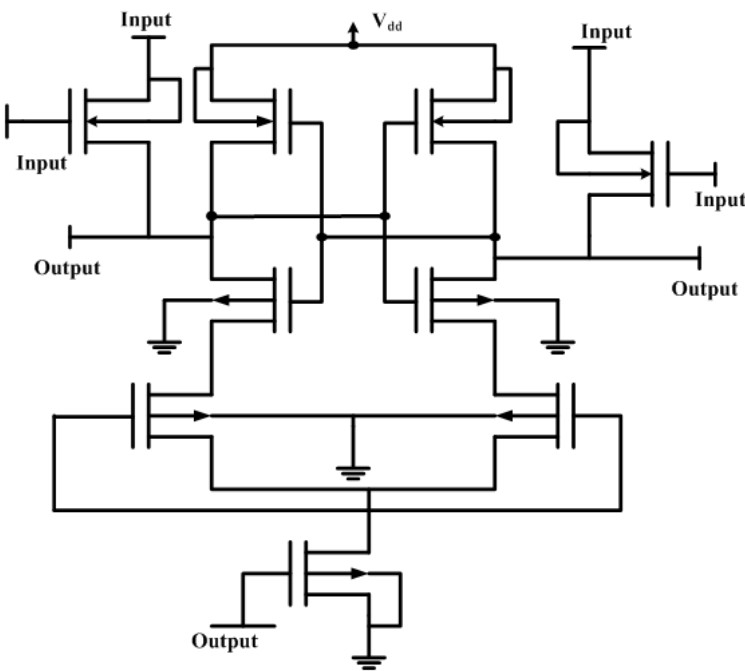

**Figure 6.** Sense amplifier.

### 4.4. Write Driver and Tri-State Logic

Before or during the active state of the word lines of the selected cell, the write driver is responsible for releasing one of the bit lines from the precharge level to a level below the write margin. The write driver, which fully swing discharges the bit line from the precharge level to the ground, is typically started by the write enable (WE) signal. The word line and write drivers can be enabled in any sequence, and this has no effect on the write process.

By including a third state, high-impedance logic, the tri-state logic expands the fundamental 1 and 0 logic states that a port can be in integrated devices. In this high-impedance state, the port is essentially erased from the circuit, as if it never existed. As an outcome, the port does not appear to exist in the third state of high impedance such that the high-impedance state acts as a selector, isolating dormant circuits. Several integrated circuits, including microprocessors, RAM or memory, and numerous chips used within device drivers, internally employ tri-state logic. Many of them are governed by active-low input, which determines whether the output leads or pins should be in a high-impedance state or drive their loads, that is to output the conventional 1 or 0.

### 4.5. 6T Single-Bit Circuit

A 6T SRAM single-bit circuit is the circuit made up of a combination of the 6T SRAM bit cell, write driver, sense amplifier, and precharge circuit, which is depicted in Figure 7. A 6T SRAM cell contains a bit and bit' pair, commonly referred to as a bit line pair. At the end of each bit line, a PMOS is wired to the precharge circuit. The precharge input is zero when the bit lines in this circuit are properly equalised and precharged to $V_{dd}$ using an equaliser transistor. Three PMOS transistors make up a precharge and equaliser circuit, and the bit lines are connected to $V_{dd}$ when the transistors are in the ON state, or when the precharge circuit is active-low.

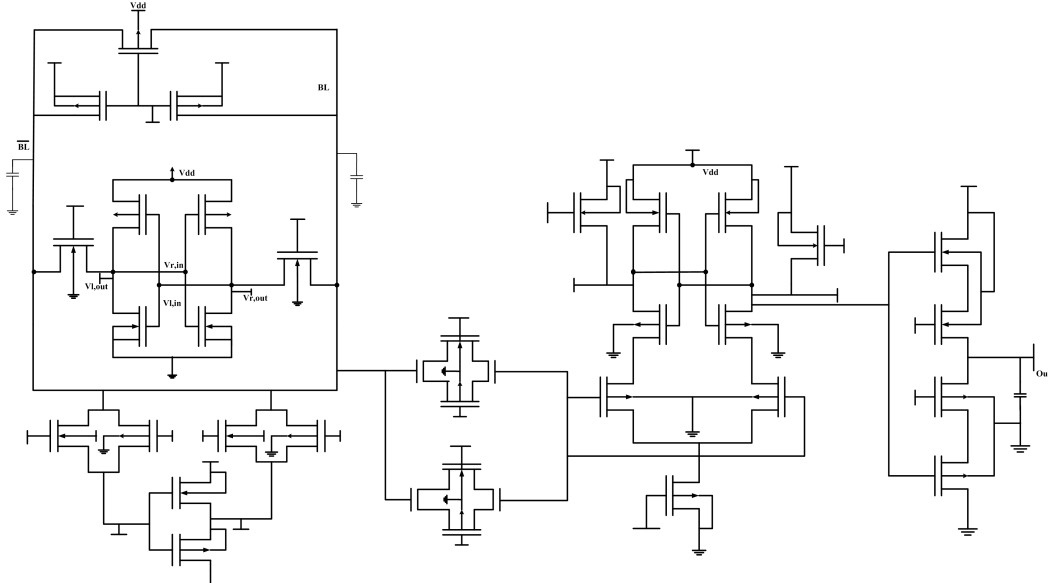

**Figure 7.** 6T single-bit circuit.

A small analogue differential voltage that forms the bit lines during a read access cycle is amplified by the sense amplifier (SA). The amplification generates a single-ended digital output with full-swing. Because the driver transistors do not have to fully discharge the bit lines when utilising the SA, the SRAM cell is smaller. The SA has to meet a few electrical requirements in order to function effectively. The least differential voltage swing required at the SA's input should, to begin with, be lower than the minimal differential voltage created across the bit lines by the SRAM cell. After receiving a minimal input differential voltage, the SA should be able to deliver the output within the sense amplification time.

This SRAM offers read and write properties.

### 4.5.1. Read Operation

During the read operation in the 6T SRAM, both capacitors are precharged to $V_{dd}$ using the precharge circuit. The word line is set to high, as well as read enable. Sense enable is also set to high. Assume that Q is set to "0". On the left part of the bit cell, we have $V_{ds} = V_{dd}$, i.e., there is some current flow in that part. The charge in the capacitor decreases little by little. Both bit lines are taken and sent to the sense amplifier; here, the sense amplifier acts as the comparator, and the bit line is shown as "0" in the output. If Q is set to be "1", then the voltage in the right section of the bit cell decreases, and the output shown will be "1". Here, the sense amplifier amplifies the output by restricting all the noise caused.

### 4.5.2. Write Operation

During the write operation in the 6T SRAM, the bit line is precharged to $V_{dd}$ and $\overline{BL}$ is left floating. Assume that either "1" or "0" is stored in Q. The *BL* and $\overline{BL}$ are used as the input lines. Write enable is set to high, and read enable is set to low. Data are given through the write driver. They pass along the bit because WE is set to 1, as the WL is set to high. The voltage in the bit decreases, which leads to turning off the n-left. That implies the p-left is on. This means that Q obtains the value of $V_{dd}$, which is 1. Here, the initial value of Q is "1" or "0", but finally, it is rewritten to be "1".

In the schematic of the 6T-SRAM with a 32 × 32 memory array, each row has the same word line, that is the first row of the schematic will have the word line as WL = 1. Therefore, when the data are sent there, that row is only activated with write enable set to high. When the data are read, the read enable signal is set to high with write enable set to low. To control which row should store the data and from which column the data should be read,

we designed decoders, which act as the controller. Both decoders are 5 × 32 decoders, each for selecting a row and a column.

## 5. Result Analysis

The bit cell used to build a 1 KB (32 × 32) array was a 6T SRAM, and it is crucial to take into account all the characteristics connected to the bit cells because they are the essential element in creating a memory array. The 6T SRAM array design parameters are illustrated in Table 1, and the dimensional parameters of the CMOS technology are depicted in Table 2.

**Table 1.** SRAM array design parameters.

| Design Parameters | Value |
|---|---|
| Technology | 22 nm CMOS |
| SRAM Array | 32 × 32 |
| SRAM Bit Cell | 6T SRAM |
| Supply Voltage ($V_{dd}$) | 0.6 Volts |

**Table 2.** Dimensional parameters of CMOS technology.

| Device Technology: CMOS | | | |
|---|---|---|---|
| Parameters | PMOS | NMOS | Access Transistors |
| Channel Length (nm) | 22 | 22 | 22 |
| Channel Width (nm) | 22 | 44 | 28 |

The graphs were plotted for the write and read access and for power. For the write operation, the $BL$, BLB ($\overline{BL}$), Q, Qb($\overline{Q}$), WL, precharge, data, Datab ($\overline{Data}$), write enable (WE), and source voltages were plotted. For the read operation, the $BL$, WL, RE, and precharge were plotted. For access, the WL was given as the pulse input, and for power, the WL was given as the PWL (linear) from 0 to $V_{dd}$, i.e., 1 was taken. The precharge was taken as $V_{dd}$ in all cases. For the write operation, WE was set to high and RE was set to low, and vice versa for the read operation.

For the power and source currents plotted on a graph, the average of that plot was calculated using the in-built calculator in the software. Both negative and positive current averages were added to obtain the total average current. This average current was then multiplied by $V_{dd}$ to obtain the total power.

### 5.1. Simulation of 6T Single-Bit Circuit

Write access, write power, read access, and read power were calculated for the 6T single-bit circuit.

Here, WE was set high. The data were given a random pulse. When the WL and WE were high, then the data were written in Q, or else, this was set to low. The WL is given as the pulse signal. The write access and write power of the 6T single-bit circuit are depicted in Figures 8 and 9. The read access and read power of the 6T single-bit circuit are depicted in Figures 10 and 11.

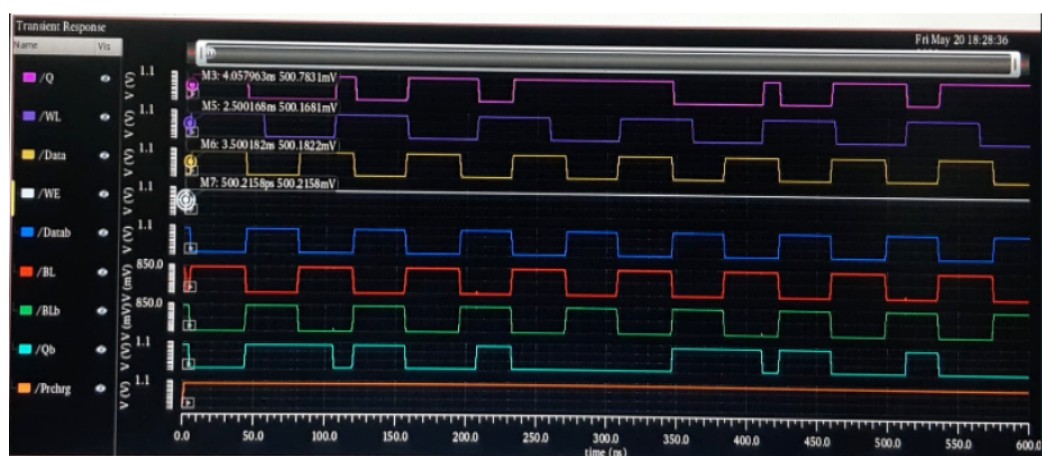

**Figure 8.** Write access for 6T single-bit circuit.

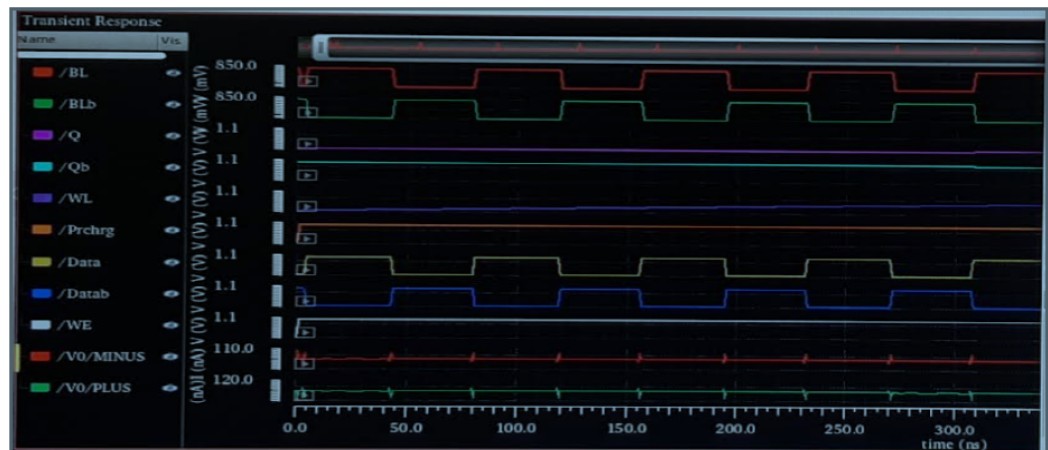

**Figure 9.** Write power for 6T single-bit circuit.

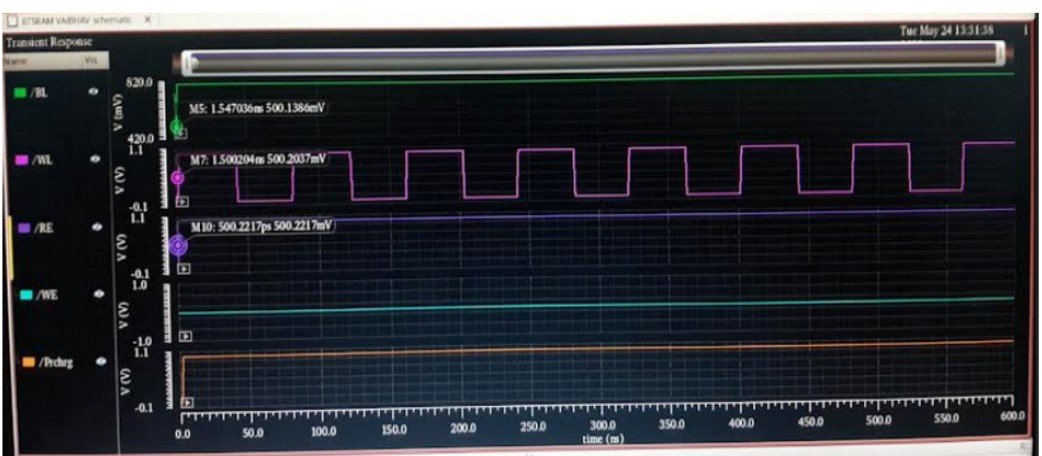

**Figure 10.** Read access for 6T single-bit circuit.

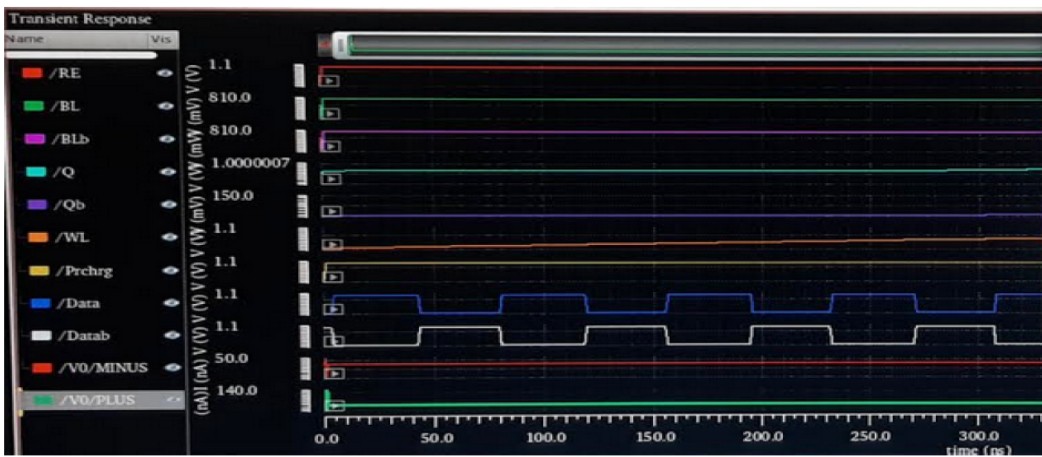

**Figure 11.** Read power for 6T single-bit circuit.

The write and read access delays for the 6T single-bit circuit are clearly depicted in Table 3.

**Table 3.** Write and read access delays for the 6T single-bit circuit.

|  | Output Node | Delays (ns) |
| --- | --- | --- |
| Write Access | Q-WL | 1.43 |
|  | Q-Data | 0.53 |
|  | Q-WE | 3.65 |
| Read Access | BL-WL | 0.14 |
|  | BL-RE | 0.97 |

Static power dissipation is the amount of power lost as heat while the circuit is operating in static mode, which happens when there is a direct path between $V_{dd}$ and the ground. This suggested methodology focuses on designing an SRAM array with a CMOS-based internal architecture in which the circumstance where both the PMOS and NMOS are on simultaneously for a very short length of time occurs. The static power dissipation of the 6T SRAM is clearly displayed in the Table 4 below.

**Table 4.** Static power analysis parameters of 6T SRAM.

| Design Parameters | Value |
| --- | --- |
| Leakage current | 18.65 pA |
| Leakage power | 187 pW |
| Transient power | 35.78 nW |
| Static power dissipation | 11.19 pW |

*5.2. Simulation of 32 × 32 Array*

The write access and write power of the 32 × 32 array circuit are depicted in Figures 12 and 13. The read access and read power of the 32 × 32 array circuit are depicted in Figures 14 and 15.

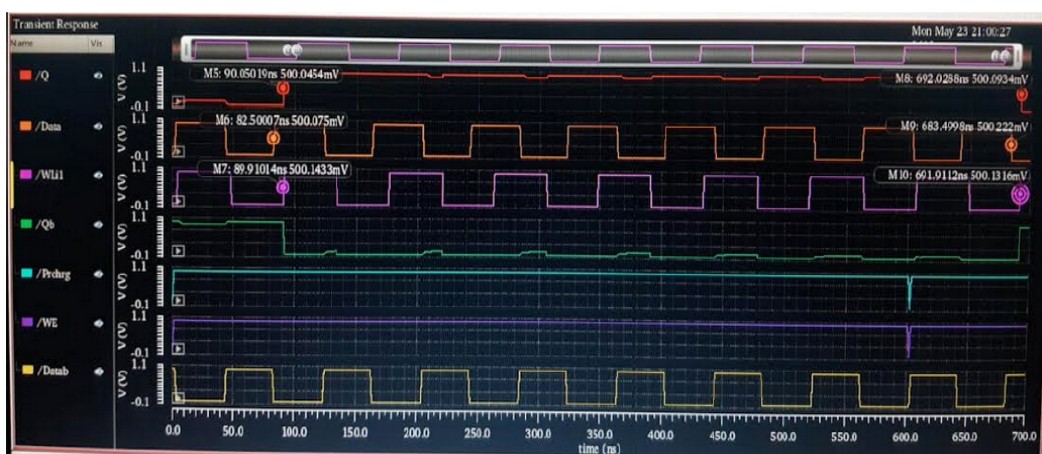

**Figure 12.** Write access for the 32 × 32 array.

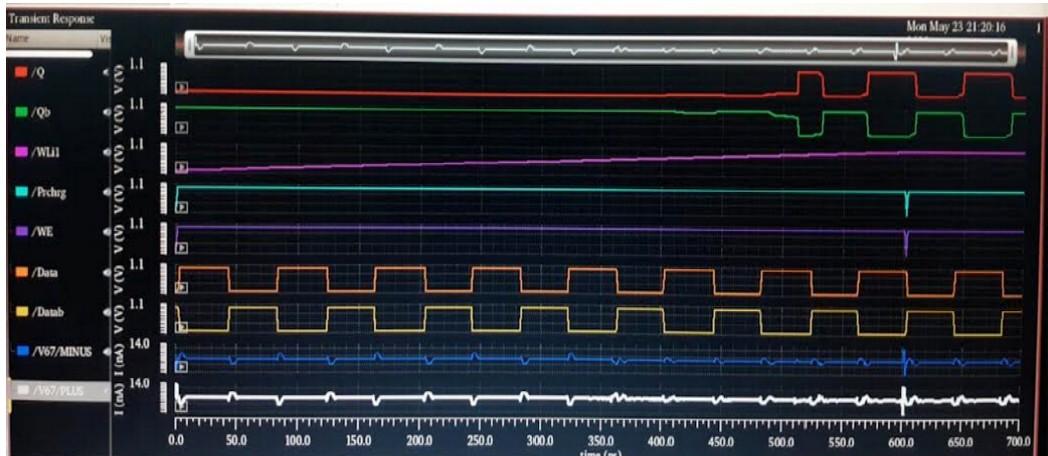

**Figure 13.** Write power for the 32 × 32 array.

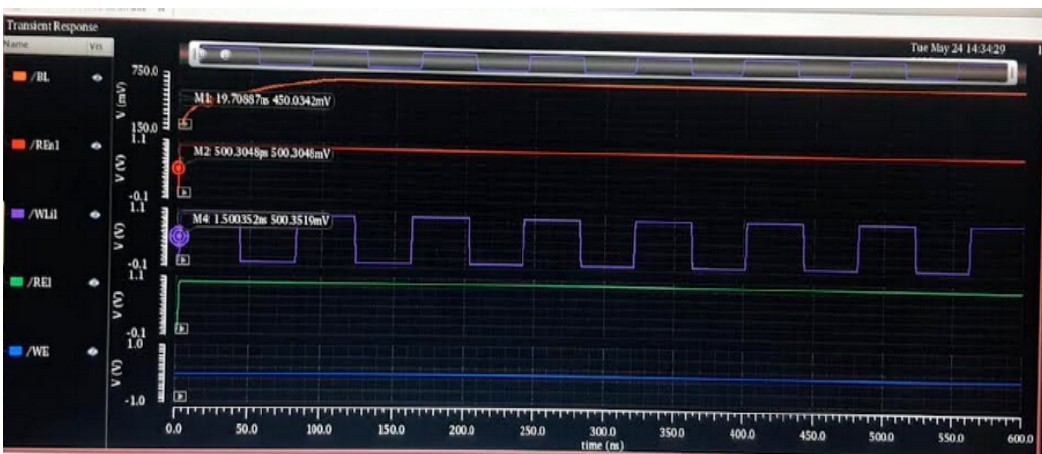

**Figure 14.** Read access for the 32 × 32 array.

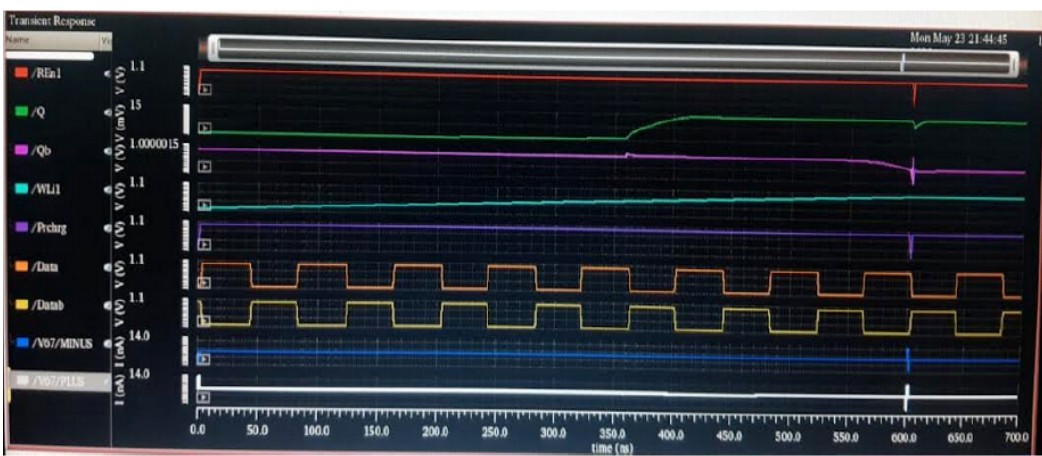

**Figure 15.** Read power for the 32 × 32 array.

The write and read access delays for the 32 × 32 array are clearly depicted in Table 5.

**Table 5.** Write and read access delays for 32 × 32 array.

|  | Output Node | Delays (ns) |
|---|---|---|
| Write 1 Delay Access | Q-WL | 7.15 |
|  | Q-Data | 2.65 |
|  | Q-WE | 21.9 |
| Write 0 Delay Access | Q-WL | 8.45 |
|  | Q-Data | 8.44 |
|  | Q-WE | 45.76 |
| Read Access | BL-WL | 7.65 |
|  | BL-RE | 11.67 |

The comparison of the power for the read and write operation in the 32 × 32 SRAM array is depicted in Table 6. The power consumption of the proposed 1 KB memory array using the proposed 6T SRAM cell in 45 nm technology for read and write operations is 50.46 µW and 410 µW, respectively, The power consumption of the proposed 1 KB memory array using the proposed 6T SRAM cell in 22 nm technology for read and write operations is 48.22 µW and 385 µW, respectively. Hence, our 6T SRAM approach outperformed in power consumption when compared to the existing 8T SRAM [21] and 7T SRAM [23] methods.

**Table 6.** Comparative analysis for power consumption of 32 × 32 SRAM array.

| Power Consumption (µW) | | | | |
|---|---|---|---|---|
| Parameters | 8T SRAM [21] | 7T SRAM [23] | Proposed 6T SRAM | Proposed 6T SRAM |
|  | (45 nm) | (45 nm) | (45 nm) | (22 nm) |
| Read Operation | 67.68 | 51.57 | 50.46 | 48.22 |
| Write Operation | 512 | 447 | 410 | 385 |

## 6. Conclusions

In this research paper, a 1 KB SRAM array in bit orientation with a 6T SRAM cell in CMOS technology was designed and analysed. It features a 1024-bit capacity for storage. The proposed 1 KB SRAM memory array took into account the minimal power usage, low

leakage, and compact feature sizes. After accounting for all the factors, it was observed that there was a substantial difference in the power consumption of the 1 KB-SRAM-based memory during the read and write operations. A 6T SRAM cell with a minimum leakage current of 18.65 pA and an average delay of 19 ns was used to implement the array structure. The power consumption for read and write operations was 48.22 μW and 385 μW, respectively, and used for the write driver circuit for low-power applications. The performance attributes of the CMOS-based SRAM arrays, such as the power dissipation for the read and write operations, were contrasted with those of previous works. This paper was validated using the Cadence Virtuoso tool in CMOS 22 nm technology.

**Author Contributions:** Conceptualization, A.S.K.; methodology, A.S.K. and M.V.S.V.; software, A.S.K.; A.S., T.D. and M.V.S.V.; validation, A.S.K.; formal analysis, X.X., O.I.K. and G.M.A.; data curation, R.P.S. and X.X.; Writing-original draft, R.P.S.; writing—review and editing, X.X., O.I.K. and A.S.K.; supervision, R.P.S. and G.M.A.; project administration, O.I.K.; funding acquisition, X.X. All authors have read and agreed to the published version of the manuscript.

**Funding:** This research work was supported by Fujian Provincial Key Laboratory of Big Data Mining and Applications, Fujian University of Technology, China.

**Data Availability Statement:** Not applicable.

**Acknowledgments:** This work was supported by the National Natural Science Foundation of China (No. 62172095) and the Natural Science Foundation of Fujian Province (Nos. 2020J01875 anPd 2022J01644).

**Conflicts of Interest:** The authors declare no conflict of interest.

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
