# Peer review of "Design and Performance Analysis of 32 × 32 Memory Array SRAM for Low-Power Applications"

_electronics, doi:10.3390/electronics12040834_

Round 1
Reviewer 1 Report
Please see the file attached.

Author Response
Respected Professor,
We are very much thankful for reviewing the manuscript and giving valuable comments on our research paper. Please find the revised version of our manuscript “Design and Performance Analysis of 32 x 32 Memory array SRAM for Low-Power Applications”, which we would like to submit as a revision to the “MDPI-Electronics Journal”.
The comments of the editor/ reviewers were greatly helpful and enabled us to improve the quality of our manuscript. We have made the necessary corrections in the revised manuscript according to the suggestions given by the reviewers. We are uploading (a) our point-by-point response to the comments (below) (response to reviewers), and (b) an updated manuscript with yellow highlighting indicating changes.
Thanks and Best Regards!
Yours Sincerely,
Dr. Aruru Sai Kumar,
Corresponding Author.
Reviewer 2 Report
This manuscript's title is "Design and Performance Analysis of 32 x 32 Memory array SRAM for Low-Power Applications," which is a great interesting topic for its area and definitely for the researchers. This manuscript will be a great source if there are some updates.
Some of my comments;
1- Line 243# In the line, has been mentioned the graphs. What are the charts? Are the authors saying the software-created screenshots?
2- Figures 8 to 15 need to recapture. They are not clear to see and not aligned in the graphs. It looks like the authors didn't spend time getting the results' snapshots.
3- Figure 2 should be moved before section 4.
4- Figures 8 and 9; figures 10 and 11, 12 and 13; and 14 and 15 should be combined due to related results.
5- All figures should have more definitions to describe the graphs or results.
6- How did the authors implement the cells in figures 1 to 7?
7- How did you apply the conditions on the cadence virtuoso tool with 22 nm technology?
After these concerns, I believe this manuscript may reach out to more audiences.
Thanks
Author Response

(The authors gave the same response as above.)

Reviewer 3 Report
The first 4 lines of the abstract can be removed. It is also preferable if the contribution is replaced with research objectives in the Introduction section since the contribution is mentioned in the conclusion section. It is preferable if section 3 is merged with section 4. Section 5 can be System Simulation and Evaluation with more details about how the proposed system is simulated. Finally, it is preferable if a subsection within section 5 is included to assess the results obtined.
Author Response

(The authors gave the same response as above.)

Round 2
Reviewer 1 Report
Thanks for the revised version. Most of my components have been addressed somehow. Please see below minor other comments.
1. GENERAL COMMENTS
Explicitily write in the paper the version used of Cadence Virtuoso used.
2. TYPOGRAPHICAL AND EDITORIAL
- 64: please rephrase “in order to continuous development”
- 75: cadence virtuoso -> Cadence Virtuoso
- 76: change colon (:) with period (.)
- 117: demonistrated -> demonstrated
Author Response
Ms. Amber Huang,
Assistant Editor,
MDPI-Electronics Journal.
Respected Editor,
Ref.: Manuscript ID: electronics-2149630.
We are very much thankful for reviewing the manuscript and giving valuable comments on our research paper. Please find the revised version of our manuscript “Design and Performance Analysis of 32 x 32 Memory array SRAM for Low-Power Applications”, which we would like to submit as a revision to the “MDPI-Electronics Journal”.
The comments of the editor/ reviewers were greatly helpful and enabled us to improve the quality of our manuscript. We have made the necessary corrections in the revised manuscript according to the suggestions given by the reviewers. We are uploading (a) our point-by-point response to the comments (below) (response to reviewers), and (b) an updated manuscript with yellow highlighting indicating changes.
Thanks and Best Regards!
Yours Sincerely,
Dr. Aruru Sai Kumar,
Corresponding Author.
Response by the Authors:
Response to Reviewer #1 (Round-2)
Thanks to the reviewer for the valuable suggestions.
- GENERAL COMMENTS
1) Explicitily write in the paper the version used of Cadence Virtuoso used.
Author response and action: Thank you for your suggestion. In the revision, we have modified the manuscript as per the reviewer's suggestions and highlighted it with yellow colour.
- TYPOGRAPHICAL AND EDITORIAL
-64:please rephrase “in order to continuous development”
- 75: cadence virtuoso -> Cadence Virtuoso
- 76: change colon (:) with period (.)
- 117: demonistrated -> demonstrated
Author response and action: Thank you for your suggestion. In the revision, we have modified the manuscript as per the reviewer suggestions and highlighted with yellow colour.